# Daphne: Multi-Pass Compilation of Probabilistic Programs into Graphical Models and Neural Networks

**Christian Weilbach**                                                       *weilbach@cs.ubc.ca*
*Department of Computer Science*
*University of British Columbia*

**Frank Wood**                                                              *fwood@cs.ubc.ca*
*Department of Computer Science*
*University of British Columbia*

**Reviewed on OpenReview:** *https://openreview.net/forum?id=XXXX*

## Abstract

Daphne is a probabilistic programming system that provides an expressive syntax to denote a large, but restricted, class of probabilistic models. Programs written in the Daphne language can be compiled into a general graph data structure of a corresponding probabilistic graphical model with simple link functions that can easily be implemented in a wide range of programming environments. Alternatively Daphne can also further compile such a graphical model into understandable and vectorized PyTorch code that can be used to train neural networks for inference. The Daphne compiler is structured in a layered multi-pass compiler framework that allows independent and easy extension of the syntax by adding additional passes. It leverages extensive partial evaluation to reduce all syntax extensions to the graphical model at compile time.

## 1 Introduction

Probabilistic modeling is fundamental to modern machine learning and statistics. The recent rise of generative AI has placed many deep learning applications within an approximate Bayesian modeling framework (Brown et al., 2020; Rombach et al., 2022). At the same time, structured probabilistic programming systems are rapidly gaining traction in statistics (Štrumbelj et al., 2024). These systems allow users to specify models and inference problems in programming languages that balance expressivity and tractability. While Turing-complete probabilistic programming languages offer universal expressivity, they can make inference intractable. In contrast, more restricted languages can still capture large classes of problems while enabling more efficient inference.

An important class of probabilistic programming languages are those that compile to representations optimized for powerful inference algorithms, such as Stan (Carpenter et al., 2017). These systems transform an expressive input syntax into a lower-level model that can be evaluated efficiently using the numerical primitives of an inference engine. Additionally, an ideal probabilistic programming system should allow users to extend its syntax and should be embedded in a rich programming environment that includes both probabilistic modeling primitives and modern machine learning tools. This combination of properties makes such a system particularly well suited for teaching and research.

Following these design principles, we introduce Daphne.[1] Among the probabilistic programming systems we are aware of, Daphne stands out due to its expressive higher-order syntax and its extensive use of partial evaluation in its compiler (Section 4).

---

[1] https://github.com/plai-group/daphne

## 2 Motivation for Daphne

To demonstrate the use cases of Daphne, we highlight two applications in Figure 1 and Figure 2. The implementation details of these papers are not covered here, they are shown to highlight the benefit of having Daphne as an inference problem specification interface to succinctly describe probabilistic graphical models. The graphical models are then used in both cases to sparsify neural network architectures for amortized inference learning tasks. The language is designed to facilitate extensions that extract rich, yet easy-to-process, compute graphs from probabilistic programs for such applications.

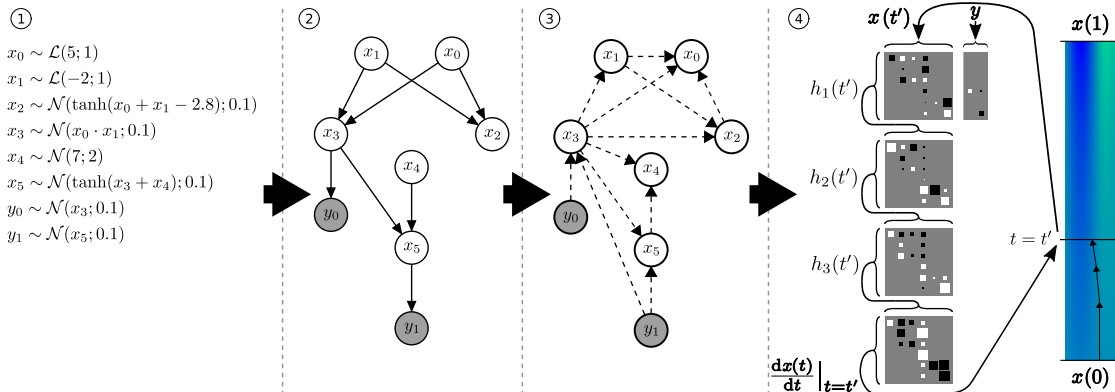

Figure 1: Translation from a Daphne program (here written in statistical syntax) into its compute graph, its faithful inverse, and then into a sparsity mask of a multilayer perceptron (MLP) guiding a continuous normalizing flow (Grathwohl et al., 2018). The resulting sparsified MLP can be trained and executed more efficiently. Figure taken from (Weilbach et al., 2020).

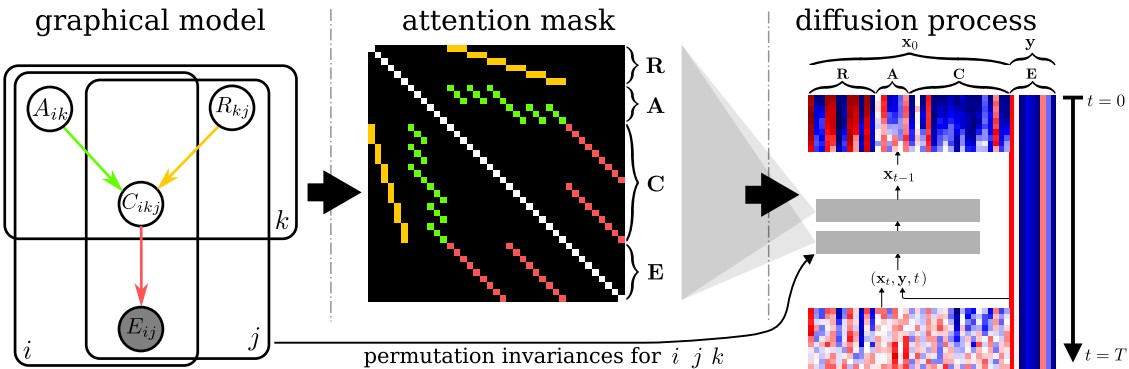

Figure 2: Integration of a graphical model extracted from Daphne into a transformer-based diffusion model trained for amortized inference tasks. On the left, the graphical model is extracted from Daphne (or manually specified), its adjacency matrix structures the sparse self-attention transformer layers guiding a diffusion process. Figure taken from (Weilbach et al., 2023b).

Daphne provides these graphical model translations in a modular translation process that eases adjustments to the specification language, and to the properties that can be captured. As an example of an extension that would be easy to add to Daphne, the permutation invariances projected in Figure 2 could also be extracted automatically, since they are statically induced by independence of link functions from plate indices (Weilbach et al., 2023b).[2] Both papers require the translation to a graphical model before they are trained with traditional parametric variational families. The ability of Daphne to reduce an expressive language to a simple data structure of a graphical model is particularly useful for such integrations.

---

[2]This was not done for this paper since the neural network was directly implemented in PyTorch.

# 3 Graphical Probabilistic Programming Language

A *graphical* probabilistic programming language (GPPL) is a language in which all random variables can be identified without evaluating the program's stochastic expressions—Stan (Carpenter et al., 2017) being a prominent example. Such a language enables loops and control flow to be unrolled at compile time, explicitly representing all random variables in a traditional probabilistic graphical model (PGM) (Koller & Friedman, 2009).

This contrasts with general probabilistic programming languages (PPLs), which support loops and recursion constructs that cannot be simplified before execution—e.g., when a loop or recursion termination condition depends on a random variable sample. van de Meent et al. (2018) distinguish these as first-order (FOPPL) vs. higher-order (HOPPL) languages. In their formulation, FOPPL enforces termination guarantees by requiring bounded loops and prohibiting higher-order functions and recursion to avoid unbounded computation. We found these restrictions unnecessarily limiting, leading us to develop GPPLs. Nonetheless, FOPPL remains a valid subset of our GPPL (Section 5).

## 3.1 Syntax

The core language syntax supported by the compiler is defined by the following grammar in Backus-Naur form (BNF):

$$
\begin{aligned}
s &::= \text{symbol (indicating variables)} \\
c &::= \text{constant value or primitive operation (syntactic atoms)} \\
f &::= \text{procedure} \\
e &::= c \mid s \mid (\texttt{let } [s_1 \ e_1 \ \ldots \ s_n \ e_n] \ e_b) \mid (\texttt{if } e_1 \ e_2 \ e_3) \mid (e_f \ e_1 \ \ldots \ e_n) \\
&\quad \mid (\texttt{sample } e) \mid (\texttt{observe } e_1 \ e_2) \\
q &::= e \mid (\texttt{defn } f \ [s_1 \ \ldots \ s_n] \ e) \ q
\end{aligned}
$$

Language 1: Daphne GPPL.

The syntax follows the Clojure programming language (Hickey, 2008; 2020). For readers unfamiliar with Lisp syntax, it can be understood as an extended form of JSON, where "code is data." Lisp expressions use a list-based meta-syntax, explicitly grouped with parentheses—i.e., expressions begin with "(" and end with ")". As seen in the grammar rule for $e$, expressions can contain symbols, making them *symbolic expressions (sexps)*.

The language supports *lexical binding* through the `let` form and *control flow* via conditional expressions using `if`. Function applications follow *prefix notation*, where the function appears first, followed by its arguments, e.g., (`+ 1 2`). Here, $f$ must evaluate to either a previously defined procedure or a primitive operation.

Functions can be defined using `defn`, enabling code reuse via named procedures. The last expression in a program serves as the *global entry point*, providing access to all defined functions. Like Anglican (Tolpin et al., 2016), Daphne includes *probabilistic primitives*:

- `sample` to draw from a random variable

- `observe` to condition the variable on data.

## 3.2 Example: Bayesian Linear Regression

A Bayesian linear regression example in Daphne is shown in Program 2. A normal prior is assigned to `slope` and `bias`, and the function `reduce` iterates over six $(x, y)$ data pairs, applying the normal likelihood function `observe-data` before returning the posterior parameters. Unlike van de Meent et al. (2018), Daphne allows recursion and higher-order functions, as demonstrated in our simple implementation of `reduce`.

```
(defn reduce [f acc s]
  (if (> (count s) 0)
    (reduce f (f acc (first s)) (rest s))
    acc))

(defn observe-data [acc data]
  (let [slope (first acc)
        bias  (second acc)
        xn    (first data)
        yn    (second data)
        zn    (+ (* slope xn) bias)]
    (observe (normal zn 1.0) yn)
    [slope bias]))

;; global entry point
(let [slope (sample (normal 0.0 10.0))
      bias  (sample (normal 0.0 10.0))
      data  [[1.0 2.1] [2.0 3.9] [3.0 5.3]
             [4.0 7.7] [5.0 10.2] [6.0 12.9]]]
  (reduce observe-data [slope bias] data)
  [slope bias])
```

Program 2: Daphne GPPL Example - Linear regression

## 4 Compiler

The *Daphne compiler* runs ahead-of-time (AOT) before inference is conducted.[3]

The compiler takes as input the full program syntax—including input data and function definitions, as seen in Program 2—and translates it into a dictionary describing a *probabilistic graphical model*. Daphne provides JSON export for its compiler passes via its command-line interface.

### 4.1 Compiler passes

The compilation process happens in multiple passes, where each pass defines the translation of specific language features into a simpler syntax before ultimately leaving only the graphical model representation (van de Meent et al., 2018). This approach facilitates the construction of extensible compilers using a layered, modular architecture that processes individual language features separately (Keep & Dybvig, 2013).

The compiler includes the following standard passes:

- `desugar`: Factorizes large `let` bindings into nested single bindings.

- `symbolic-simplify`: Applies operations on syntactic objects whenever possible. For example, (`first` [`sample1`]) simplifies to `sample1`, since the argument is a syntactically represented vector and can be evaluated symbolically.

- `partial-evaluation`: Evaluates parts of the program ahead of time, as described in Section 4.2.

- `substitute`: An explicit substitution pass that replaces symbols with values while respecting the binding structure of the language.

---

[3]Daphne can be understood as a two-level approach (Nielson & Nielson, 1992), or more generally, as a staged computation system. The first interpreter translates the GPPL into a graphical model representation, and the second interpreter evaluates the 0th-order language expressions of the graphical model during inference. For clarity, we will refer to the first interpreter as the compiler and the second as the inference runtime.

## 4.2 Partial Evaluation

Partial evaluation (Jones et al., 1993) is a technique that interprets parts of a program ahead of time, effectively implementing compilation by running the interpreter. It evaluates sub-expressions as soon as sufficient information is available. For example, `(+ 1 2)` can be evaluated to `3` and replaced in the program code since all necessary information is available.

The Daphne language, like Clojure (Hickey, 2020), is a functional, stateless Lisp, naturally supporting substitution semantics, which is particularly well-suited for partial evaluation (Jones et al., 1993).

Daphne applies partial evaluation using a *fixed-point operator* alongside other compiler passes, continuing iteration until no further simplifications are possible. It follows Clojure's `eval` semantics, traversing sub-expressions in a top-down manner, using a properly scoped environment, until a sub-expression can be fully evaluated. Code that does not depend on random variables can be fully reduced by the partial evaluator, while expressions involving random variables are simplified as much as possible (van de Meent et al., 2018), as shown in the compiled graph for linear regression in Program 3.

Since fixed-point iteration is used, the compiler does *not* have a constant number of passes, and it is non-trivial to determine in advance how many iterations will be required. Furthermore, functions defined using `defn` naturally allow *bounded recursion* during partial evaluation, provided that their inputs shrink at each recursive step.[4]

## 4.3 Graphical model

After iteratively substituting, expanding and reducing the input syntax a graph data structure with vertices `:V`, adjacency `:A`, link functions `:P` and observed nodes `:Y` remain,

```
{:V #{sample1 sample2 observe3 observe4 observe5 observe6 observe7 observe8},
 :A
 {sample2 #{observe3 observe4 observe5 observe6 observe7 observe8},
  sample1 #{observe3 observe4 observe5 observe6 observe7 observe8}},
 :P
 {sample1 (sample* (normal 0.0 10.0)),
  sample2 (sample* (normal 0.0 10.0)),
  observe3 (observe* (normal (+ (* sample1 1.0) sample2) 1.0) 2.1),
  observe4 (observe* (normal (+ (* sample1 2.0) sample2) 1.0) 3.9),
  observe5 (observe* (normal (+ (* sample1 3.0) sample2) 1.0) 5.3),
  observe6 (observe* (normal (+ (* sample1 4.0) sample2) 1.0) 7.7),
  observe7 (observe* (normal (+ (* sample1 5.0) sample2) 1.0) 10.2),
  observe8 (observe* (normal (+ (* sample1 6.0) sample2) 1.0) 12.9)},
 :Y
 {observe3 2.1, observe4 3.9,  observe5 5.3,
  observe6 7.7, observe7 10.2, observe8 12.9}}
```

Program 3: Compiled Graphical Model - Linear regression

The link functions of `:P` are expressed in a language that requires neither variable binding nor loop support. For example, `(observe* (normal (+ (* sample1 1.0) sample2) 1.0) 2.1)` represents an observation statement in its simplified form. This representation can be readily evaluated using NumPy or PyTorch arithmetic and probability primitives in Python, facilitating implementations such as ancestral sampling (Appendix A.1).

Many compilers transform code into A-normal form (ANF) or single static assignment (SSA), which closely resemble this graphical representation. As a result, Daphne can map these computations efficiently to low-level languages that lack garbage collection, particularly CUDA.

---

[4]This condition is currently not enforced, meaning the compiler may not terminate if violated. Termination checks could be implemented in the partial evaluator, similar to runtime contracts in Racket (Nguyen et al., 2019), but would provide guarantees at translation time in this context.

In correspondence with the syntax primitives of the GPPL language, `observe*` and `sample*` refer to low-level implementations of *sampling* and *log-probability evaluations*, respectively, ensuring computational efficiency in probabilistic inference.

### 4.4 Inference runtime

Daphne also provides optional runtime support for inference. A differentiable subset of the GPPL is supported with source to source reverse-mode automatic differentiation (Baydin et al., 2017) and can be plugged into a simple Hamiltonian Monte Carlo (HMC) implementation. A core set of derivatives such as those for normal distributions, arithmetic and a set of scalar functions is provided, but might require additional definitions of derivatives for wider classes of programs. There is also support for Metropolis within Gibbs sampling. Both of these implementations have been used for testing and teaching and provide a starting point for further exploration, but should not yet be expected to perform competitively with mature probabilistic programming systems. Daphne builds on well tested Anglican (Tolpin et al., 2016) primitives though and and can be extended to a wide range of MCMC methods if needed.[5]

Research with Daphne led to exploration of variational inference methods for amortized inference (Weilbach et al., 2020). In this work the compiler furthermore provides a sparse inversion of the graphical model structure according to Webb et al. (2018) and a translation of the graphical model to Python code. This additional compilation step provides a human readable PyTorch implementation of sampling and log probability evaluation of prior and likelihood for a given graphical model. The code supports batching and can be used to sample a synthetic data set for training a continuous normalizing flow (Grathwohl et al., 2018) as described in (Weilbach et al., 2020). Follow-up work has extended this approach and leveraged the diffusion model framework to yield reliable and more scalable amortized inference artifacts (Weilbach et al., 2023b;a). Daphne can provide automatic derivation of the attention masks from the GPPL language for the sparse transformer in this line of work (Weilbach et al., 2023b) as shown in Section 2.

## 5 Deep learning primitives

$$e ::= \ldots \mid (\texttt{foreach}\ e_c\ [s_1\ e_1\ \ldots\ s_n\ e_n]\ e'_1\ \ldots\ e'_m) \mid (\texttt{loop}\ e_c\ e_{init}\ f_{acc}\ e_1\ \ldots\ e_n)$$

Language 2: Loop extensions.

Daphne also optionally supports the two loop forms of van de Meent et al. (2018). These forms do not make the language more expressive, but have been originally used to implement a linear algebra library for (Weilbach et al., 2020), which includes matrix multiplication and 2d convolution. Both `foreach` and `loop` require a loop counter $e_c$ expression evaluating to an integer to determine the number of loop iterations. `foreach` binds $s_1, \ldots, s_n$ with each element of the collection yielding expressions $e_1, \ldots, e_n$ in the body expressions $e'_1, \ldots, e'_m$. `loop` iterates an accumulating function $f_{init}$ starting with $e_{init}$ over $e_1, \ldots, e_n$. To illustrate the use of the language we provide an excerpt of the library together with an example 2d convolution invocation at the end.

```
(defn dot-helper [t state a b]
  (+ state
     (* (get a t)
        (get b t))))

(defn dot [a b]
  (loop (count a) 0 dot-helper a b))

(defn row-mul [t state m v]
  (conj state (dot (get m t) v)))
```

---

[5]There is also zero-copy access to all of Python through `https://github.com/clj-python/libpython-clj` or `https://github.com/oracle/graalpython`

```clojure
(defn mmul [m v]
  (loop (count m) [] row-mul m v))

(defn row-helper [i sum a b]
  (+ sum
     (dot (get a i)
          (get b i))))

(defn inner-square [a b]
  (loop (count a) 0 row-helper a b))

(defn inner-cubic [a b]
  (apply + (foreach (count a) [n (range (count a))]
                    (inner-square (get a n) (get b n)))))

(defn slice-square [input size stride i j]
  (foreach size [k (range (* i stride)
                          (+ size (* i stride)))]
           (subvec (get input k)
                   (* j stride)
                   (+ size (* j stride)))))

(defn slice-cubic [inputs size stride i j]
  (foreach (count inputs) [input inputs]
           (slice-square input size stride i j)))

(defn conv-kernel [inputs kernel bias stride]
  (let [ic (count (first inputs))
        size (count (first kernel))
        remainder (- size stride)
        to-cover (- ic remainder)
        iters (int (Math/floor (/ to-cover stride)))]
    (foreach iters [i (range iters)]
             (foreach iters [j (range iters)]
                      (inner-cubic (slice-cubic inputs size stride i j)
                                   kernel)))))

(defn conv2d [inputs kernels bias stride]
  (foreach (count kernels) [ksi (range (count kernels))]
           (conv-kernel inputs (get kernels ksi) (get bias ksi) stride)))

(let [w1 [[[0.8, 0.9],
           [0.9, 0.6]],
          [[0.0, 0.0],
           [0.1, 0.5]]],
         [[[0.2, 0.4],
           [0.6, 0.1]],
          [[0.4, 0.5],
           [0.1, 0.1]]]]
      b1 [0.1, 0.2]
      x  [[[0.4, 0.5, 0.8, 0.8]
           [0.5, 0.8, 0.6, 0.1]
           [0.9, 0.4, 0.7, 0.2]
           [0.5, 0.0, 0.4, 0.2]],
          [[0.0, 0.8, 0.2, 0.3]
           [0.2, 0.2, 0.8, 0.7]
           [0.1, 0.6, 0.6, 0.3]
           [0.6, 0.7, 0.5, 0.2]]]]
  (conv2d x w1 b1 2))
```

In combination those primitives can be used to implement the deep learning applications in (Weilbach et al., 2020), including a stochastic deconvolution layer and a small convolutional network.

## 6 Related work

Table 1: Comparison of Probabilistic Programming Systems

| Feature | Daphne | Stan | PyMC | Anglican | Pyro | Gen |
|---|---|---|---|---|---|---|
| Turing-complete | ✓ | ✗ | ✓ | ✓ | ✓ | ✓ |
| Stochastic recursion | ✗ | ✗ | ✓ | ✓ | ✓ | ✓ |
| Graphical model focus | ✓ | ✓ | ✗ | ✗ | ✗ | ✗ |
| Higher-order functions | ✓ | ✗ | ✗ | ✓ | ✓ | ✓ |
| Recursion support | ✓ | ✗ | ✓ | ✓ | ✓ | ✓ |
| HMC inference | (✓) | ✓ | ✓ | ✗ | ✓ | ✓ |
| Python integration | ✓ | ✓ | ✓ | ✗ | ✓ | ✗ |
| Compilation to CUDA | ✓ | ✗ | ✗ | ✗ | ✗ | ✗ |

Compared to Turing-complete languages such as Anglican (Tolpin et al., 2016), Pyro (Bingham et al., 2019), PyMC (Oriol et al., 2023), or Gen (Cusumano-Towner et al., 2019), the Daphne language only supports programs without stochastic recursion. This restriction prevents it from implementing certain non-parametric models and complex stochastic recursion schemes. However, inference in such models can be highly challenging and brittle, making the restriction to graphical models an acceptable trade-off for many applications.

While Daphne shares most of its syntax with Anglican, a key difference is that Anglican programs cannot be translated into a graphical model ahead of inference time and integrated into other inference systems, such as the deep learning pipelines shown in Section 2. Daphne, by contrast, is designed to first compile into a graphical model before inference.[6] Anglican, on the other hand, can freely execute any Clojure code within its inference runtime and provides a set of powerful Monte Carlo sampling-based inference methods.

Stan (Carpenter et al., 2017) is a well-established GPPL designed to work seamlessly with its Hamiltonian Monte Carlo (HMC) inference engine. Compared to Stan, Daphne offers a significantly more expressive functional probabilistic programming syntax, including support for recursion and higher-order functions. BUGS, which has a similar syntax to Stan, has been translated into the FOPPL language in (van de Meent et al., 2018). Likewise, the Daphne graphical model output could be translated back into Stan to leverage its inference engine while benefiting from Daphne's more expressive syntax.

Unlike Stan, which is implemented in C++, Daphne is built in Clojure—a high-level, interactive programming environment naturally suited for meta-programming and rapid prototyping. Daphne's codebase is small and modular, with compiler passes and core semantics implemented in at most a few hundred lines of self-contained code. This makes Daphne highly adaptable to new ideas while providing seamless integration with the broader Clojure ecosystem, such as PyTorch through `libpython-clj`.[7] Further exploration of this runtime integration is left for future work.

In terms of user experience, Stan provides more extensive documentation and a polished interface, making it more accessible to users who do not wish to modify the system. Daphne, by contrast, is designed for flexibility and extensibility, prioritizing expressivity and compiler-level transformations over ease of use.

## 7 Conclusion and future work

Daphne is a small yet versatile probabilistic programming environment that represents a new point in the design space of probabilistic programming languages and compilers. Its goal is to capture as much of higher-

---

[6]There are also some exploratory interpreters included in the code base for teaching purposes, but they are not the main focus of the system.

[7]https://github.com/clj-python/libpython-clj

order functional programming as possible while remaining reducible to traditional probabilistic graphical models ahead of inference time. The representations produced at different compiler stages can be exported to provide fine-grained program information, such as dependencies between random variables, for downstream inference runtimes. These representations have been used in a line of research on structured neural networks for amortized inference and for teaching graduate courses in probabilistic programming.

**Limitations**  For very large input programs, such as deep learning models, partial evaluation can lead to expression swell, significantly slowing down compilation as all linear algebra operations need to be syntactically expanded and reduced. Scaling better to such programs requires further research, particularly in structuring the expansion and reduction of expressions during partial evaluation to ensure that reductions are always applied before expansions. Many other probabilistic programming systems (Bingham et al., 2019; Tolpin et al., 2016) treat tensors as single objects rather than explicitly modeling the full computational graph, including all scalars and intermediate computations, as Daphne does. While this avoids the problem, it comes at the cost of not tracking fine-grained sparse structures.

Beyond these improvements, expression swell is inherently unavoidable in some cases, as Daphne unrolls all branches and loops into a graph before execution. Consider a Markov Decision Process (MDP), such as an agent navigating a maze. Each additional time step into the future requires expanding all branches with all possible subsequent branches, leading to an exponential blow-up in the representation. This makes GPPL-based systems like Daphne unsuitable for such problems.

Adding a module system, likely following Clojure's approach, is left for future work. Currently, the linear algebra functions in Section 5 must be manually concatenated with the program before compilation. Additionally, the Daphne language does not yet support anonymous function definitions, which would enable closures, making it more in line with standard Clojure and more convenient for complex programs. Supporting closures is planned for a future iteration of the language.

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

# A    Appendix

## A.1    Python export

The following export is done with the help of hy-lang[8] which allows a direct translation between lisps and then into Python syntax. This is the compiled code for Program 2.

```python
import hy
import torch
import math
from torch.distributions import Normal, Bernoulli, Laplace, Uniform

class Model:
    dim_latent = 2
    dim_condition = 6
    faithful_adjacency = [[0, 1], [0, 2], [0, 3], [0, 4], [0, 5], [0, 6], [0, 7], [1, 2], [1, 3], [1, 4],
                          [1, 5], [1, 6], [1, 7], [0, 0], [1, 1]]
    src = '((defn\n  reduce\n  [f acc s]\n  (if (> (count s) 0) (reduce f (f acc (first s)) (rest s)) acc))\n (defn\n
observe—data\n  [acc data]\n  (let\n   [slope\n    (first acc)\n    bias\n    (second acc)\n    xn\n    (first data)\n
yn\n    (second data)\n    zn\n    (+ (* slope xn) bias)]\n   (observe (normal zn 1.0) yn)\n   [slope bias]))\n (let\n
[slope (sample (normal 0.0 10.0)) bias (sample (normal 0.0 10.0))]\n  (reduce\n   observe—data\n   [slope bias]\n
[[1.0 2.1] [2.0 3.9] [3.0 5.3] [4.0 7.7] [5.0 10.2] [6.0 12.9]]))\n  [slope bias]))\n'

    def sample(self):
        sample_1 = Normal(0.0, 10.0).sample()
        sample_0 = Normal(0.0, 10.0).sample()
        observe_4 = Normal(sample_0 * 3.0 + sample_1, 1.0).sample()
        observe_6 = Normal(sample_0 * 5.0 + sample_1, 1.0).sample()
        observe_5 = Normal(sample_0 * 4.0 + sample_1, 1.0).sample()
        observe_7 = Normal(sample_0 * 6.0 + sample_1, 1.0).sample()
        observe_2 = Normal(sample_0 * 1.0 + sample_1, 1.0).sample()
        observe_3 = Normal(sample_0 * 2.0 + sample_1, 1.0).sample()
        return [torch.tensor([sample_0, sample_1]),
                torch.tensor([observe_2, observe_3, observe_4, observe_5, observe_6, observe_7])]

    def log_likelihood(self, sample, observe):
        log_likeli = torch.zeros(sample.shape[0])
        log_likeli += Normal(sample[[slice(None), 0]] * 2.0 + sample[[slice(None), 1]], 1.0).log_prob(observe[[slice(None), 1]])
        log_likeli += Normal(sample[[slice(None), 0]] * 5.0 + sample[[slice(None), 1]], 1.0).log_prob(observe[[slice(None), 4]])
        log_likeli += Normal(sample[[slice(None), 0]] * 6.0 + sample[[slice(None), 1]], 1.0).log_prob(observe[[slice(None), 5]])
        log_likeli += Normal(sample[[slice(None), 0]] * 1.0 + sample[[slice(None), 1]], 1.0).log_prob(observe[[slice(None), 0]])
        log_likeli += Normal(sample[[slice(None), 0]] * 3.0 + sample[[slice(None), 1]], 1.0).log_prob(observe[[slice(None), 2]])
        log_likeli += Normal(sample[[slice(None), 0]] * 4.0 + sample[[slice(None), 1]], 1.0).log_prob(observe[[slice(None), 3]])
        return log_likeli

    def log_prior(self, sample):
        log_prior = torch.zeros(sample.shape[0])
        log_prior += Normal(0.0, 10.0).log_prob(sample[[slice(None), 1]])
```

---

[8]https://hylang.org/

```
        log_prior += Normal(0.0, 10.0).log_prob(sample[[slice(None), 0]])
    return log_prior
```

