# OpenReview forum: "Daphne: Multi-Pass Compilation of Probabilistic Programs into Graphical Models and Neural Networks"
_TMLR — Accepted by TMLR_

### Review · Reviewer_2Cmz · 2024-10-06

**Summary Of Contributions:**

In this paper, the authors introduce Daphne - a higher-order probabilistic programming language that can be used to define probabilistic graphical models with support for higher-order functions and an unbounded number of random variables. The contributions can be summarized as follows:
1. Programs written in Daphne can be directly compiled into graphical models using their multi-pass compiler
2. Unlike most traditional and well-known probabilistic programming languages, Daphne supports higher-order functions and recursions and can directly convert these into structured probabilistic models at the compilation step
3. Daphne programs can also be compiled into deep-learning models for use with PyTorch
4. Models defined using Daphne have limited support for inference but can be compiled to Stan and take advantage of the latter's mature inference pipeline

**Audience:**

Yes

**Claims And Evidence:**

Yes

**Requested Changes:**

The paper is in pretty good shape as it is. The only thing I would recommend is what I mentioned in the Weaknesses section. It would be great to see a more in-depth comparison with other well-known probabilistic programming languages (particularly the higher-order ones). A table would be best, but a couple of paragraphs highlighting the differences should also be sufficient.

**Strengths And Weaknesses:**

Strengths:
1. The paper is well-written and discusses both the strengths and limitations of Daphne in substantial detail
2. The paper is filled with helpful examples that do a wonderful job of explaining the basics of Daphne syntax and its usage in different contexts
3. The Daphne language itself appears to be very expressive and would definitely be useful in the context of both teaching and research
4. The compiled graphical model output by the model appears to be translatable to a wide variety of formats (including Stan and PyTorch), which is a huge plus

Weaknesses:
The only real weakness I see here is a lack of discussion of related work. It is clear from the paper what Daphne is good at and how versatile and general it is. However, I am sure that readers such as myself who have dabbled in some of the more popular probabilistic programming frameworks such as PyMC would be interested to see a more detailed comparison of how Daphne compares to these. Some of these comparisons can already be inferred from the paper such as Daphne's expressive syntax that allows for higher-order functions and recursions. However, a more in-depth comparison would certainly be interesting to see, particularly with respect to the other higher-order languages such as Pyro, etc.

---

> ### Author Response · Authors · 2024-12-07
>
> Thank you for your encouraging review. We have expanded on the related work section and also clarified the limitations of Daphne more with the latest revision.

---

### Review · Reviewer_cAHD · 2024-11-26

**Summary Of Contributions:**

The manuscript presents Daphne, a probabilistic clojure-like lisp that (AOT) compiles graphical models into a symbolic representation amenable for learning and inference using existing ML backends.

**Audience:**

Yes

**Claims And Evidence:**

No

**Requested Changes:**

### Comparative Analysis / Performance

I would like to see the following:

[c1] Some detailed comparisons (both qualitative and quantitative) with Anglican, Stan, and other relevant modern probprog frameworks (Pyro?).

[c2] A few concrete examples where Daphne offers advantages over existing approaches.

[c3] Some benchmarks on compilation time, and inference efficiency for the default backend (or all available / supported backends).

### Case studies on ergonomics and features

[c4] The manuscript includes a very few details on how the language is generally run and how programs managed by the user. Nowadays I believe it is reasonable to expect papers presenting these kinds of details at least as an FYI. Particularly, one of the exciting contributions is the ability to compile the lisp down to a python representation, which makes a pytorch / jax / $FRAMEWORK translation possible. I was disappointed when the authors showed the users having to use Hy as a middle step, which makes it seem as aftertought.

[c5] The manuscript includes a couple of toy examples, which are great and clear. However, it also claims that some work used Daphne and its advantages to implement state of the are inference for interesting problems. Using these papers as case studies, particularly focusing on how Daphne enabled the implementation of the methods (which would otherwise have been challenging in other sysmtes), would greatly enhance the contribution of this manuscript.

### Positioning

[c6] The manuscript could use from committing to a specific target audience. Currently it reads well if one is faimilar with both the PL and ML literature, but some of the language seems to be too PL-coded in a way that might make the average TMLR reviewer puzzled at times. Note that it would be perfectly fine to keep it PL-targeted, but at that point it would not necessarily be a good manuscript for TMLR.

[c7] I would look to significantly strengthen the motivation for why this system is needed. If the primary contribution is the ability to represent graphical models symbolically, I would double down on that and focus on providing motivation to the reader that it is an impactful constraint.

### Assorted questions

#### Intro

>An interesting position in the language design space are languages that can be readily translated into a representation for which efficient inference algorithms exist, e.g. Stan (Carpenter et al., 2017). Such systems compile an expressive input syntax into a lower-level model that can be efficiently evaluated with the numerical primitives of the inference engine. Furthermore it should be possible for users to extend the language with new syntax if needed.

This paragraphs reads a little weirdly to me. It seems to miss an argument that justifies why translating down to graphical models is useful. The two other sentencea seem to read as a bucket list of properties that would make for a good PPL, so I wonder if the paragraph had additional language that was mistakenly hidden.

>[...] particularly well suited for teaching and research

As opposed to what? And is the intention of the work to particularly overfit to these cases (which seem very broad, especially when including a generic "research").

#### GPPL

>The last expression is the global entry point into the program having access to all defined functions.

This is a fairly long-winded and SICP-like way to say that language has an implicit global entrypoint. A quick code listing with comments would probably be more useful (and be perfectly clear).

Tha language also doesn't seem to include a module / import / load system. Is that correct? If so, how are complex programs likely to be representable / reusable in a growing experimental codebase?

>In contrast to van de Meent et al. (2018) Daphne does not rule out recursion or higher-order functions as can be seen in our simple implementation of reduce.

Is this novel wrt. Anglican / Stan / et al.? I think it's a little strange (though still defensible) to employ a teaching-focused-- daresay toy? --language as the baseline for such things.

#### Compiler

>This approach allows to build extensible compilers with towers of languages that deal with single language features

I suspect even good readers won't be familiar with this concept. Might be worth a quick explanation if this requirement greatly informed the design of Daphne.

>The need of potentially inconsistent compilation semantics separate from the interpreter

Would it be possible to include an example of failure modes of this kind? I couldn't think of one without going into fairly complex scenarios, but it would be good to understand what types of common patterns the authors are trying to mitigate away.

>Daphne also provides optional runtime support for inference. A differentiable subset of the language is supported with source to source reverse-mode automatic differentiation (Baydin et al., 2017) and can be plugged into a simple Hamiltonian Monte Carlo (HMC) implementation.

It would be great to significantly expand this section to describe the variuos inference runtimes. The ability to autodifferentiate through the language is an extremely exciting feature, but without knowing the relevant subset it is impossible for a reader to understand whether Daphne might fit their research problems.

#### Related work

>Inference in such models is often very challenging and brittle though and this motivates our restriction to graphical models.

It's not clear whether this fits as a motivation. Are the authors arguing that PPL developers should be discouraged from supporting stochastic recursion?

Note that I would understand _if_ that was indeed a reason, but it does need at least some kind of qualitative argument based on the literature or the authors' experience.

>BUGS, which is similar to Stan in syntax, has been translated to the Daphne language in van de Meent et al. (2018)

This seems to imply that Daphne was designed in van de Meent et al. 2018. If that's the case, I would reframe this paper to characterize that accurately.

#### Conclusions

>Very large sized input programs, such as deep learning models, partial evaluation can lead to expression swell that significantly slows down compilation as all linear algebra operations need to be syntactically expanded and reduced.

This is an important limitation, and a critical one, given how strongly the manuscript focuses on one of the motivations being that the language enables good use of "modern ML primitives". I think more clarity on the limitations, and guidance on follow-up work that is going to remove these seems warranted.

**Strengths And Weaknesses:**

## Strengths
[s1]: The manuscript is easy to follow if one is familiar with the literature, gives a good amount of context, and overall reads fairly well.

[s2]: Daphne itself is fairly straightforward as a language, and the paper does a good job at presenting its overview.

[s3]: the language design seems to hit a good balance between expressivity and tractability, and the manuscript includes literature that has apparently already successfully made use of the language.

## Weaknesses
[w1]: the manuscript lacks... purpose. It generally lacks a _clear_ articulation of what makes Daphne fundamentally different from existing PPLs. In particular, I think a 1:1 comparison with Anglican is necessary, given the clear (and stated) similarity in both design and (perhaps) capabilities.

[w2]: the manuscript is at times confusing wrt. its origin, and what is specifically novel about the contribution. Is Daphne fundamentally a formalization of the language presented in de Meent et al., 2018? Is it the AOT symbolic compiler? Is it presenting some novel _and qualitatively better_ ergonomics point in design space from a user perspective?

[w3]: while the limitation section is welcomed, it is still really limited for the critical learning and inference cases where the complexity of the model becomes non-trivial. Since the manuscript doesn't go into details in _how_ any of it scales (e.g. how long the AOT steps take as the graphical model gets bigger, or as more higher order functions are used), a reader may justifiably label the language as a useful teaching tool, rather than a powerful system to do bayesian inference research with.

---

> ### Author Response · Authors · 2024-12-07
>
> Thank you for your comprehensive feedback. We hope that all your concerns are sufficiently addressed with the revision (see the breakdown there). We are happy to add more details if needed.

---

> > ### Comment · Reviewer_cAHD · 2024-12-17
> > **Thank you for the update!**
> >
> > Thank you for the update! I'll take a look at the revision.
> >
> > For future submissions:
> > - it would be great if such significant changes were highlighted in a way that makes it easy to take a diff between versions (say--using a different color for the new text).
> > - I would have appreciated a breakdown with pointers wrt. my review, especially given the many questions I had... :-)

---

### Review · Reviewer_ADBH · 2024-11-27

**Summary Of Contributions:**

This paper introduces Daphne, a probabilistic programming system that supports the representation and compilation into graphical models and (simple and small) neural networks. The submission describes the language syntax, compiler design, and example code to represent linear regression and convolutional neural networks.

**Audience:**

No

**Broader Impact Concerns:**

No concerns.

**Claims And Evidence:**

No

**Requested Changes:**

1. The submission needs to clearly explain what are the compelling use cases of the proposed system, and provide more details on what are the new features that are not supported by other probabilistic programming systems.

2. The authors also need to expand the related work section. From the current submission, it is unclear why designing another probabilistic programming system for representing neural networks is necessary.

**Strengths And Weaknesses:**

The direction of designing probabilistic programs to represent neural networks and study inference methods is interesting.

However, with the current form of the submission, it is hard to capture why it is beneficial to use the proposed Daphne system, and what are the unique advantages compared to other probabilistic programming systems.

In particular, I am not sure whether the topic of this paper is a good fit for TMLR. It is important to clarify what are the key takeaways for audience working on deep neural networks to learn from this work.

---

> ### Author Response · Authors · 2024-12-07
>
> Thank you for your feedback. We have added two use cases of the system as a new Section 2 and have also expanded the related work section with the revision. The two use cases emphasize the benefits the system had for building probabilistic deep learning systems, if there are additional details that the reviewers are interested in we are happy to expand further. The actual work is covered in two published ML papers.

---

### Decision · Action_Editor_6xXz · 2025-02-11

**Recommendation:** Accept as is

**Comment:**

The authors introduce and provide a high-level overview of Daphne, a probabilistic programming system with a fairly expressive syntax which is still sufficiently restrictive to allow for effective sampling-based inference. This is an appealing combination which occupies an under-explored niche in the probabilistic programming space. All the reviewers recommended acceptance as the latest revision addressed most of their concerns.

**Audience:**

Yes

**Claims And Evidence:**

Yes